# Inversion of Soil Heavy Metal Content Based on Spectral Characteristics of Peach Trees

**Wei Liu**, **Qiang Yu** *, **Teng Niu, Linzhe Yang and Hongjun Liu**

College of Forestry, Beijing Forestry University, Beijing 100083, China; vivian_liu@bjfu.edu.cn (W.L.);
niuteng21@bjfu.edu.cn (T.N.); ylz15235456201@163.com (L.Y.); liuhongjun_keai9@bjfu.edu.cn (H.L.)
* Correspondence: yuqiang@bjfu.edu.cn

**Abstract:** There exists serious heavy metal contamination of agricultural soils in China. It is not only time- and labor-intensive to monitor soil contamination, but it also has limited scope when using conventional chemical methods. However, the method of the heavy metal monitoring of soil based on vegetation hyperspectral technology can break through the vegetation barrier and obtain the heavy metal content quickly over large areas. This paper discusses a highly accurate method for predicting the soil heavy metal content using hyperspectral techniques. We collected leaf hyperspectral data outdoors, and also collected soil samples to obtain heavy metal content data using chemical analysis. The prediction model for heavy metal content was developed using a difference spectral index, which was not highly satisfactory. Subsequently, the five factors that have a strong influence on the content of heavy metals were analyzed to determine multiple regression models for the elements As, Pb, and Cd. The results showed that the multiple regression model could better estimate the heavy metal content with stable fitting that has high prediction accuracy compared with the linear model. The results of this research provide a scientific basis and technical support for the hyperspectral inversion of the soil heavy metal content.

**Keywords:** fresh peach leaf; spectral index; soil heavy metal; multiple-regression model; inversion





## 1. Introduction

Soil is the basis for plant growth and development, supplying the water, air, and mineral elements on which plants depend. At the same time, it is also the basis for building a good ecological environment and an important factor in maintaining national ecological security. In 2014, China conducted a national survey on soil pollution, which showed an alarming overall soil environment, especially for heavy metal pollution on farmland and industrial and mining wasteland [1,2]. With the rapid development of industry, agriculture, and urbanization, environmental issues have gradually been brought to people's attention, among which air and water pollution have become prominent and drawn great attention. However, soil pollution has not received sufficient attention due to its lagging nature and hidden nature [3,4]. Although soil has a certain ability to purify pollutants, when the amount of soil pollutants exceeds the environmental capacity of the soil, it will cause changes in the structure and functionality of the soil and even cause serious harm to the entire ecosystem [5]. Excessive heavy metals in soil will not only cause stress to crops and affect their growth and development [6–8], but also accumulate in crops through the food chain and ultimately in the human body, which makes them a major safety hazard to human health and the environment [9–11]. For example, Minamata disease and bone pain disease were identified to be caused by heavy metal pollution in Japan in the 1950s, which made heavy metal one of the most urgent problems in soil pollution and caused widespread concern in the international community [12,13]. Therefore, it is crucial to detect the distribution of heavy metal elements in soil in large areas rapidly, and many scholars have been trying to monitor the soil environment using hyperspectral techniques.

The traditional methods of monitoring heavy metals in soil are usually based on chemical analysis; the results are more accurate, but it requires a lot of manpower, materials, and financial resources, and the monitoring of large areas cannot be achieved [14]. In contrast, hyperspectral remote sensing technology is quicker and more efficient, has high resolution and accuracy, which makes it more suitable for monitoring at large spatial scales [15–17]. It is a promising alternative detection technology and is gradually developing and maturing [18–20].

At present, many scholars have conducted research on heavy metal pollution in soil by using ground object reflectance spectroscopy, aiming to reflect the status of soil pollution in study areas through the variation characteristics of reflectance spectroscopy [21]. Hang et al. explored the feasibility of using soil reflectance spectroscopy to estimate Cd, Pb, As, Cr, Cu, and Zn in suburban soils, and showed that the combination of visible and near-infrared reflectance spectroscopy with partial least squares regression was an alternative method for the rapid monitoring of heavy metal pollution in suburban soils [22]. Shi et al. studied the spectral variation characteristics of navel orange leaves by conducting pot experiments with artificially added cadmium (Cd), and then established a prediction model based on the spectral index and obtained good model accuracy [23]. Kooistra et al. used the hyperspectral vegetation index and red edge position to monitor the pollution status of heavy metals in floodplain soil, indicating that the spectral characteristic parameters of heavy metals in some bands could be used as effective indicators [24]. Zhao et al. selected two Cd gradient sample plots in Guixi to study the effect of rice Cd pollution on the spectral characteristics, and the results showed that with an increased Cd content in the soil, the depth of the blue and red valleys became shallow, the slope of the red pass became smaller, the $NDVI_{705}$ (Normalized Difference Vegetation Index) value decreased, and the reflectance decreased in the near-infrared waveband ($R_{750-1250}$) [25].

Based on previous studies, the hyperspectral monitoring of the soil heavy metal content can be divided into the following two aspects: the direct estimation of heavy metal content using soil reflectance spectroscopy [26–28] and the indirect estimation of heavy metal content using vegetation reflectance spectroscopy [29,30]. The direct estimation method was established by finding a direct relationship between the concentration of heavy metals in soil and the spectral reflectance, but in fact, most of the surface soil is covered by vegetation, which makes it difficult to obtain the soil reflectance spectra directly, and that also leads to difficulties in data acquisition and processing. The indirect estimation method tries to excavate the correlation between the elemental content of soil heavy metals and the vegetation reflectance spectra in order to establish an inversion model. It remains to be further studied how to make full use of the characteristics of heavy metals in soil, fully exploit the leaf spectral information, and improve the accuracy and wide applicability of hyperspectral inversion models. In addition, pot experiments are used to artificially and quantitatively add isogradient heavy metals to study the changes in plant spectral characteristics. It allows for a better estimation of the concentration of the soil heavy metal content to be made, which is due to favorable measurement conditions. Virtually, the leaf spectra measured in the field environment are influenced by many factors, such as soil type, soil moisture, solar illumination, and ambient temperature. It is significantly different from the corresponding factors affecting the spectra obtained in the laboratory, and hence the inversion models obtained in the laboratory are often difficult to be directly applied in the field. It is necessary to establish a suitable and accurate prediction model for the heavy metal content in the field using vegetation spectroscopy. Currently, many inversion models have been developed, among which the widely used ones are linear models as well as nonlinear models. Nonetheless, these models have their own drawbacks. For example, linear models cannot fit nonlinear data, support vector machines are difficult to achieve large training samples, and random forests are prone to overfitting and difficult to maintain. In addition, the physical and chemical soil properties vary among different types of soils, as well as the variability of geographic location and environment in the study area, which will affect the prediction accuracy of the models. However, it does not mean that

the modeling has greater uncertainty; instead, it is essential for us to build corresponding prediction models for different geographical areas.

To address the shortcomings of the current study, we used peach trees as the research object to estimate the soil heavy metal element content by comparing different regression models and using leaf spectral reflectance data, which provides a new theoretical basis and innovative ideas to explore the feasibility of monitoring the soil heavy metal element concentration using hyperspectral technology and using it for spatial inversion.

## 2. Materials and Methods

### 2.1. Study Area

The study area is located in the northeastern part of Beijing, China, about 70 km from the center of the city, near a tailings pond with serious pollution (Figure 1). The region's terrain is high in the northeast and low in the southwest, with a temperate continental monsoon climate, hot and rainy in summer and cool and humid in autumn. The annual average temperature is 11.7 °C, the coldest in January with an average temperature of −5.4 °C and the hottest in July with an average temperature of 26.1 °C. The annual precipitation is 629.4 mm, mainly concentrated in summer, at 453.0 mm, which is 72% of the annual precipitation. It is rich in agricultural resources and an important agricultural and sideline product base in Beijing. It has a geographical environment that is surrounded by mountains in the north, east, and west, leading to large mountainous and semi-mountainous areas suitable for fruit and forest production. There are 720,000 mu of forest, and the ratio of economic to ecological forest is 2:3. Peach trees are widely planted in this area, and the Pinggu big peach is famous in the market and is a Chinese geographical indication product. The area contains a variety of minerals such as gold, silver, copper, and manganese. At present, there are two tailings ponds, including Liujiadian and Wanzhuang. With the intensification of agricultural production, the development of industry, the scale of livestock and poultry farms, and increased municipal waste emissions, the risk of soil heavy metal pollution in the study area has increased; therefore, it is especially urgent to be able to monitor the soil in the whole area quickly and on a large scale.

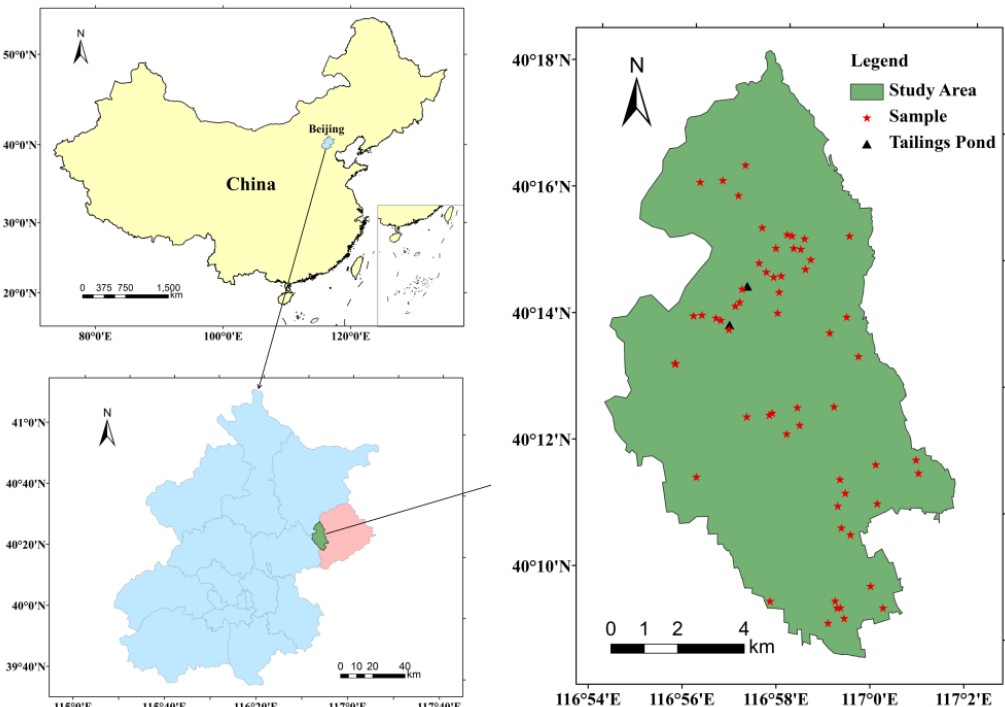

**Figure 1.** Study area and distribution of sampling points.

## 2.2. Data Acquisition

2.2.1. Peach Leaf Reflectance Spectrum Acquisition

In this study, spectra were collected using a portable FieldSpec 4 feature spectrometer, manufactured by ASD, USA. The specific operating parameters of the spectrometer are as follows: wavelength range from visible to near infrared (350–2500 nm), spectral sampling interval of 1.4 nm in the range of 350–1000 nm and 2 nm in the range of 1001–2500 nm, and a field of view of 30°. Due to the large noise generated at both ends of the spectral band, the data from 350 to 400 nm and 2400 to 2500 nm were excluded. Accordingly, a total of 2000 spectral bands of data were obtained.

First, after the survey of the study area, a reasonable survey route was formulated indoors based on the survey results. We adopted a combination of administrative and environmental units and set up 58 sampling points (Figure 1). Based on the distribution of peach tree cultivation and the information provided by farmers about peach tree growth, similar agricultural land with peach trees was selected to obtain a reliable sampling based on the administrative village, taking into account environmental factors such as the size of the peach tree cultivation area and soil type, in order to cover the variability of the study sites as much as possible. We used the 10 m × 10 m regular grid method to lay out plots at the identified sampling sites, covering the cinnamon soil and the peach trees. After that, we measured the data between 10:00–14:00 in clear weather. Along the diagonal position of the square sample plot, at the two top corners as well as at the geometric center, we, respectively, selected a mature peach tree 5–8 years old, which satisfied a height of 5–6 m and a diameter at breast height of 8–10 cm. At the same time, it records, in detail, the geographical coordinates, ambient temperature and humidity, solar radiation, etc. After the peach trees were identified, we could start measuring the leaf spectra by selecting 5–10 leaves per tree and collecting 10 spectral data points per leaf. We selected mature dark green peach leaves and required intact, healthy, and disease-free leaves. Then, we retrieved the reflectance spectrum information of peach leaves at sampling points, and the acquisition steps were as follows (Figure 2): Step 1: optimize the spectrometer, then scan the standard white plate for calibration, and start measuring when it collects a straight horizontal footprint for a certain period of time; Step 2: adjust the projection mode, then make sure the probe is 5–10 cm vertically down from the leaf surface; Step 3: utilize the instrument to collect data, and take care to keep the value stable after reading. During the collection of spectral information, we had to re-optimize the calibration every 15 min. We imported the obtained leaf spectrum curve into ViewSpec Pro (version 6.0) software and calculated the first derivative after removing the abnormal spectrum curve (Curves with apparently too high or apparently too low reflectivity). Finally, the average value of all data was calculated as the actual reflectance information of this point.

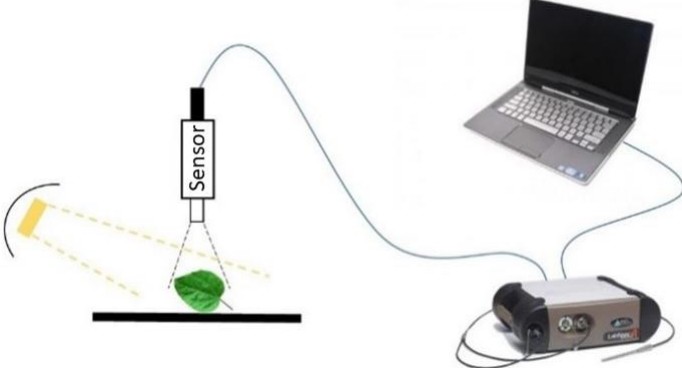

**Figure 2.** Schematic diagram of spectrometer measuring leaf spectrum.

2.2.2. Soil Sample Collection and Heavy Metal Content Determination

We collected soil samples while measuring the leaf spectrum information, and the soil type of the sample in this experiment was mainly cinnamon soil. Each sampling point

comprised 5 to 10 sub-sampling points, and the sampling depth was 0–40 cm. When taking soil samples from one sampling point, we collected samples from multiple points in the vicinity, picked off impurities such as dead leaves, root stumps, stones, and gravel, and fully mixed the samples from these sub-sampling points. The excess soil was discarded according to the quartile method, and about 1 kg was retained as a test sample for analysis and testing. The soil sample was loaded into a numbered self-sealing bag and brought back to the laboratory. After natural air-drying, the collected soil samples were passed through a 2-millimeter nylon sieve to remove impurities. Then, the samples were dried at 60 °C in an oven, ground with agate mortar, and passed through a 100-mesh nylon sieve, the result of which was sealed in a numbered bag for the next step of testing. The related data of heavy metal content in soil samples were obtained using laboratory chemical detection and analysis. The soil PH value is determined by using the spot method through a PH meter. The soil organic matter content was tested using the potassium dichromate volumetric method through an electric furnace. The element cadmium (Cd) was measured using graphite furnace atomic absorption spectrophotometry, with an atomic absorption spectrometer. The chromium (Cr), lead (Pb), and copper (Cu) were analyzed using flame atomic absorption spectrophotometry, with an atomic absorption spectrometer; the amount of total mercury (Hg) was obtained using the cold atomic absorption method with a mercury measuring instrument; and the arsenic (As) content was quantified using silver diethyldithiocarbamate spectrophotometric method with a spectrophotometer.

### 2.3. Construction of Spectral Index

In remote sensing of vegetation, the spectral index has always been regarded as an effective indicator that can monitor or evaluate growth and development [31]. It can amplify the characteristic information of the plant spectrum and reduce the interference of external factors such as the atmosphere and underlying surfaces on the spectral information. In order to extract the wavelength combination with the strongest correlation between heavy metal content and the spectral index, four typical spectral indices were selected for calculation in this study. In the 400–2400 nm spectral wavelength range, we calculated the spectral index for all wavelengths of the original hyperspectral combination in pairs. The four spectral indices were calculated as shown in Table 1.

**Table 1.** Spectral index and calculation formulas.

| Name of Spectral Index | Formula |
|---|---|
| Difference vegetation index (DVI) | $R_{\lambda_1} - R_{\lambda_2}$ |
| Simple ratio vegetation index (SRVI) | $R_{\lambda_1} / R_{\lambda_2}$ |
| Normalized difference vegetation index (NDVI) | $(R_{\lambda_1} - R_{\lambda_2})/(R_{\lambda_1} + R_{\lambda_2})$ |
| Inverse difference vegetation index (IDVI) | $1/R_{\lambda_1} - 1/R_{\lambda_2}$ |

In the formulas, $\lambda_1$ and $\lambda_2$ represent the position of any wavelength in the 350–2500 nm band, and $R_{\lambda_1}$ and $R_{\lambda_2}$ represent the original spectral reflectance at the wavelength position of $\lambda_1$ and $\lambda_2$, respectively.

### 2.4. Correlation Analysis

In order to analyze the relationship between spectral indices and soil heavy metal elements, we explored the correlation between four spectral indices with different wave combinations and six soil heavy metal elements, by which we determined the sensitive wave combinations of each heavy metal element. We introduced the Pearson correlation coefficient to describe this relationship that argues for the feasibility of indirect inversion of soil heavy metal elements using hyperspectral techniques. The calculation formula of the Pearson correlation coefficient is as follows:

$$r = \frac{\sum_{i=0}^{n}(y_i - \overline{y})(x_i - \overline{x})}{\sqrt{\sum_{i=1}^{n}(y_i - \overline{y})^2 \sum_{i=0}^{n}(x_i - \overline{x})^2}}$$

In the formula, $y_i$ represents the content of heavy metals in soil at the $i$th sampling point; $x_i$ represents the difference vegetation index (DVI), simple ratio vegetation index (SRVI), normalized difference vegetation index (NDVI) or reciprocal difference vegetation index (IDVI) of the peach leaves at the $i$th sampling point; $\overline{y}$ represents the average value of the soil heavy metal content; $\overline{x}$ represents the average value of the DVI, SRVI, NDVI, or IDVI of peach leaves; and $n$ is the number of samples.

The calculation of different spectral indices in the whole waveband was realized and run in the Python programming language, which was also used for correlation analysis between the soil heavy metal content and spectral indices.

### 2.5. Modeling Method

Before modeling, the samples are divided into two sets, the training set and the test set. The training set is used to generate the prediction model, while the test set is used to examine whether the performance of the generated model is excellent. Within the identified 58 sample points, 38 samples were randomly selected as the training set for building the prediction model, and the remaining 20 samples were chosen as the test set for evaluating the prediction accuracy of the model.

### 2.5.1. Univariate Linear Regression Model

In this paper, the regression model was constructed for spectral index and the soil heavy metal content by using univariate and multiple linear regression methods. The univariate linear regression is a method to analyze the linear correlation of only a single independent variable (independent variable x and dependent variable y). When the elements affecting the dependent variable have only one major and decisive role, the model will have good prediction accuracy. The basic form is as follows:

$$Y = ax + b$$

$$b = \frac{\sum Y_i}{n} - a\frac{\sum x_i}{n}$$

$$a = \frac{n\sum x_iY_i - \sum x_i\sum Y_i}{n\sum x_i^2 - (\sum x_i)^2}$$

In the formula, $Y$ represents the dependent variable of the prediction model and $Y_i$ is the soil heavy metal content of the $i$th sampling point; $x$ represents the independent variable of the prediction model and $x_i$ is the spectral index of the $i$th sampling point; and $a$ and $b$ are the parameters of the univariate linear regression equation.

### 2.5.2. Multiple Linear Regression Model

The multiple linear regression method was first proposed by Francis Galton in the late 19th century, which was applied to model prediction at the earliest opportunity. It predicts the independent variable, y, using the optimal combination of multiple dependent variables, x, to build a linear regression model, which is a classical statistical analysis method based on the least square method. The general expression of a multivariate linear regression equation is as follows:

$$Z = \beta_0 + \beta_1X_1 + \beta_2X_2 + \cdots + \beta_kX_k + \varepsilon$$

where $Z$ is the soil heavy metal content, $X_k$ is the spectral index of the $k$th sampling point, $\beta_k$ is regression coefficient, $n$ is the number of sampling points, and $\varepsilon$ is the random error.

### 2.6. Model Accuracy Evaluation

For assessing the fitting and generalization ability of the model, the following two determinants were used for evaluation: coefficient of determination ($R^2$) and root mean square error (RMSE). $R^2$ indicates the stability of model construction and validation [32].

The closer the $R^2$ is to 1, the more stable the model is and the better the fit is. The RMSE indicates the predictive power of the model [33]. The smaller the RMSE is, the higher the predictive accuracy is. Calculations are performed according to the following equation:

$$R^2 = 1 - \frac{\sum_{i=1}^{n} \sum_{i=1}^{n} (y - \hat{y})^2}{\sum_{i=1}^{n} \sum_{i=1}^{n} (y - \overline{y})^2}$$

$$RMSE = \sqrt{\frac{1}{n} \sum_{i=1}^{n} (y - \hat{y})^2}$$

## 3. Results

### 3.1. Statistical Characteristics of Soil Heavy Metal Content

According to the laboratory test results, the organic matter content of the soil was mainly between 10 and 30 g/kg, which is neutral to weakly alkaline, with a pH value between 6 and 8. The heavy metal content in the measured soil samples was statistically analyzed, and the results are shown in Table 2. The background values in the table are quoted from the book "Monitoring and Evaluation of Environmental Quality of Agricultural Produce", published in 2013 [34]. The standards in the table are grade I and II, as specified in the National Soil Environmental Quality Standards (GB 15618-1995) [35]. It can be seen from the table that the coefficient of variation was moderate, between 10 and 100%. The mean values of all the heavy metals exceeded the background values of the soil environment in the 58 samples collected. The average values of three metals, Cr, As and Pb, are higher than the national grade I standard, and the rest are lower than the standard value. Moreover, all the metals were lower than the national grade II standard, showing that the soil environmental quality was generally passable in the area. However, from the maximum value, the content of six heavy metal elements exceeded the national level standard, and some heavy metals exceeded the secondary standard, which indicated that the soil environment in the study area was polluted within different degrees and had a certain impact on the cultivation of peach trees.

**Table 2.** Statistics of heavy metal content in soil.

| Element | Maximum (mg/kg) | Minimum (mg/kg) | Mean (mg/kg) | Coefficient of Variation (%) | Background Value (mg/kg) | National Grade I Standard (mg/kg) | National Grade II Standard (mg/kg) |
|---|---|---|---|---|---|---|---|
| Cd | 0.76 | 0.09 | 0.27 | 58.71 | 0.12 | 0.20 | 0.30 |
| Cr | 172.99 | 52.85 | 82.81 | 24.88 | 56.47 | 90.00 | 250.00 |
| Hg | 0.24 | 0.02 | 0.06 | 57.65 | 0.04 | 0.15 | 0.30 |
| As | 132.96 | 7.75 | 28.70 | 90.85 | 8.14 | 15.00 | 30.00 |
| Cu | 53.60 | 22.18 | 31.07 | 23.08 | 18.50 | 35.00 | 50.00 |
| Pb | 497.24 | 20.36 | 91.41 | 90.83 | 13.78 | 35.00 | 300.00 |

### 3.2. Spectral Response of Peach Tree Leaves to Soil Heavy Metals

In order to show the spectral response pattern of peach tree leaves to soil heavy metal elements more clearly and intuitively, the collected leaf spectral curves were grouped according to heavy metal content with background values as interruption points (Table 3). Spectral curves with heavy metal content higher than the background value were regarded as polluted areas, and spectral curves with heavy metal content lower than the background value were regarded as background areas. Then, the average values were taken, respectively, to compare and analyze the spectral curves of the peach tree leaves.

**Table 3.** Statistics of soil heavy metal content in polluted and background area.

| Heavy Metal | Cr (mg/kg) | Cu (mg/kg) | As (mg/kg) | Cd (mg/kg) | Pb (mg/kg) | Hg (mg/kg) |
|---|---|---|---|---|---|---|
| Polluted Area | 74.85 | 29.44 | 34.36 | 0.28 | 94.45 | 0.06 |
| Background Area | 54.98 | 18.04 | 8.03 | 0.11 | 10.16 | 0.03 |

The original spectral curves of the peach leaves in polluted and the background areas are shown in Figure 3a. The trend of reflectance is generally consistent in both areas. However, due to the different content of heavy metals in the soil, the response mechanisms reflected in the different wavebands are different. From the analysis of the spectral reflectance curves of the peach tree leaves, it is found that the spectral reflectance in the background areas is generally lower than in the polluted areas, and the reflectance is positively correlated with the heavy metal content in the soil. In order to visualize the difference of the original spectral curves under different contamination, we use the spectral curves of the background area minus the spectral curves of the contaminated area and plot it as Figure 3b. As shown in Figure 3b, between the wavelengths of 760 and 1300 nm, the spectral reflectance was significantly different in the background and polluted areas, with the maximum difference reaching 13.41%, indicating that the spectrum of this band is relatively sensitive to soil heavy metals.

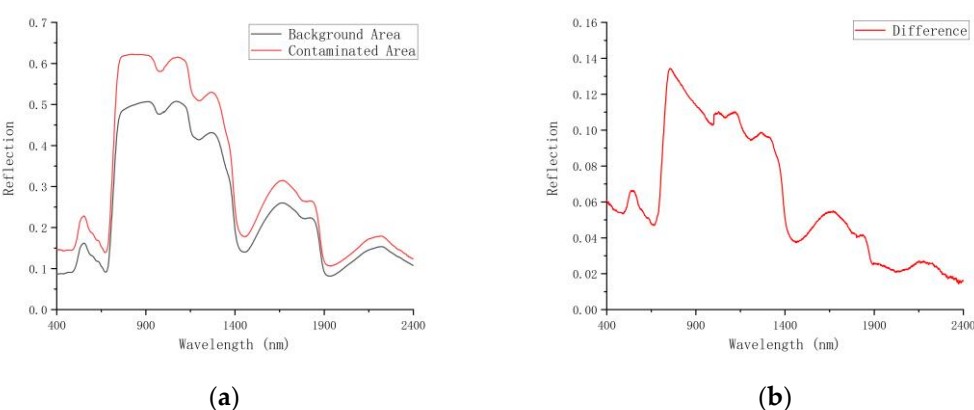

(**a**)  (**b**)

**Figure 3.** Original curves of and differences between spectra of polluted and background areas: (**a**) Original spectral curve, (**b**) Original spectral difference curve.

In the visible range of 400–760 nm, the spectral characteristics of plant leaves are mainly affected by various pigments, among which chlorophyll plays a major role. With an increased soil heavy metal content, green peak reflectance increased, with a maximum difference of 6.26%. This is because of the enrichment of heavy metals in plants, which leads to further toxicity. Furthermore, the chlorophyll content and light absorption rate decreased, resulting in increased reflectance. In the spectral range of 680–780 nm, the spectral reflectance of peach leaves increased sharply, and the typical "red edge" effect of vegetation appeared. The leaf spectral curve in the polluted areas rose significantly more than that in the background areas, and there was a high reflectance platform in the 780–1300 nm range. The reflectance of the polluted areas was between 50.96 and 61.88% and that of the background area was between 41.41 and 50.76%. There are two absorption valleys near 1460 and 1930 nm in the spectrum, which are mainly formed by water vapor absorption. The absorption valley of leaves in the polluted areas is lower than that in the background areas.

Figure 4 shows the first derivative of the peach leaf spectrum curves in the polluted and background areas. The first derivative curve can better eliminate most of the influence of background noise on the spectrum, reduce the scattering and absorption of light by the atmosphere in the process of spectrum collection [36], which can accurately determine



the position of the reflectance peak and absorption valley of the original spectrum curve. Among them, the "trilateral" parameter is important in the first derivative spectrum [37]. It can be seen from Figure 4b that the red, yellow, and blue edges of the peach tree leaves in the contaminated and background areas did not move significantly; the red edge is at 720 nm, the blue edge is at 520 nm, and the yellow edge is at 572 nm. It shows that the soil heavy metal content did not significantly interfere with the trilateral parameters. The red edge slope is the maximum value of the red edge area, which can reflect the chlorophyll content of the leaves. The analysis shows that the influence of heavy metals on the red edge slope is very obvious. With an increased soil heavy metal content, the enrichment of heavy metals in the plant leaves deepens and the chlorophyll content decreases, leading to a sharp increase in the red edge slope. The blue edge slope showed the pollution area > the background area, which increased with an increased soil heavy metal content. The slope of the yellow edge is negative in both the contaminated and background areas, and the analysis shows that the slope increases when the heavy metal contamination increases. In summary, the red, blue, and yellow edge positions of the leaves were very insensitive to the interference of soil heavy metal and showed a strong anti-interference ability. However, the slope of the edges had an obvious response to heavy metals and increased with the increased in the soil heavy metal content.

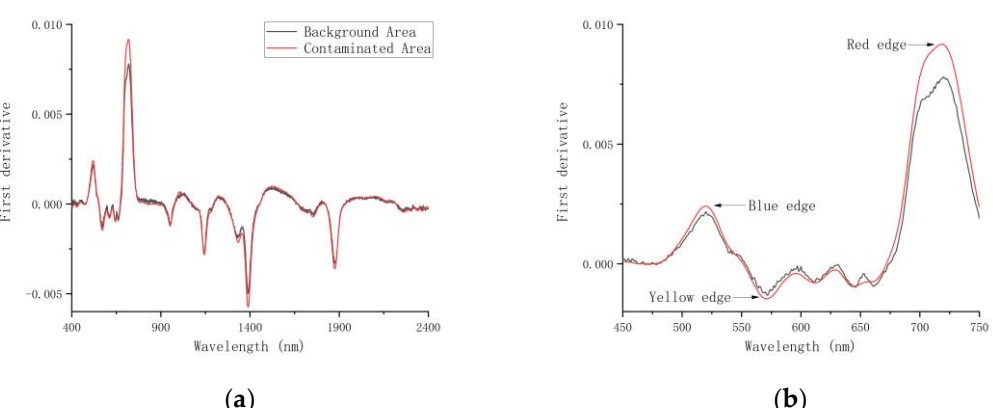

(**a**)  (**b**)

**Figure 4.** Spectral curves of the first derivative treatment of the polluted and background areas: (**a**) First derivative spectral curve; (**b**): Reflectance spectral trilateral parameter.

### 3.3. Correlation Analysis of Soil Heavy Metals and Spectral Reflectance

Figure 5 shows the correlation curve between different heavy metals in the soil and spectral reflectance. Overall, the soil Cr and spectral reflectance are moderately correlated and not significant, and the correlation coefficient value of each band is stable at about −0.6. The correlation coefficients of soil Cu and reflectance fluctuated in the range of 0.14–0.42, which revealed that there was a weak linear correlation between Cu and spectral reflectance. The correlation coefficients between Hg and reflectance are all below 0.3; therefore, it can be considered that there is no correlation between Hg and spectral reflectance. The correlation between soil As, Pb, and Cd and spectral reflectance varies greatly in different wavelength ranges, reaching the 0.1 significance level in some band ranges, and the overall correlation curve trend was the same. There is a positive correlation in the whole range, and the correlations are ranked in the order As > Pb > Cd. For example, soil As, with a strong correlation, was significantly correlated in the range of 380–515 nm, with a maximum value at 481 nm (0.76). After that, the correlation decreases and then recovers in the range of 515–674 nm, dropping to the minimum value at 553 nm (0.59). The curve is obviously concave at 674–1453 nm, and the shortwave near-infrared spectrum belongs to this range, indicating that the sensitivity of the shortwave near infrared to soil As is lower than that of the other wavebands. The variation of the long-wave near infrared is large, where all of the three elements are significantly correlated in the range of 1886–2119 nm. Meanwhile, the soil arsenic element also had a significant correlation at 2346–2400 nm.

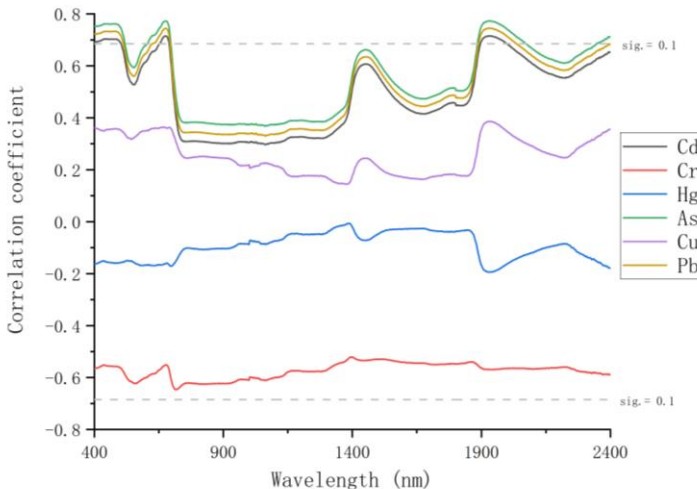

**Figure 5.** Correlation between heavy metals in soil and spectral reflectance.

### 3.4. Correlation Analysis of Soil Heavy Metals and Spectral Index

The results of the above research show that the correlation between single-band spectra and soil heavy metal elements was generally low, and only part of the band range of some elements reached a significance level of 0.1, while the other elements could be considered irrelevant. Therefore, the stability and accuracy of predicting the heavy metal content using a single band cannot meet the needs of practical application. In order to clearly characterize the sensitive bands of each element, this paper uses the data of six soil heavy metal elements to establish the spectral DVI, SRVI, NDSI, and IDVI and their correlation coefficient distribution map in the whole band. Only the heat maps of each metal and the DVI spectral index are shown here (Figures 6–11). Further information on the distribution of the correlation coefficients of each metal element with spectral indices can be found in Appendix A.

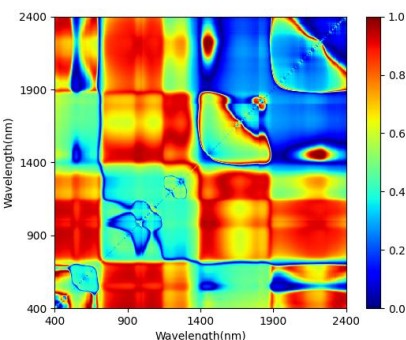

**Figure 6.** Correlation coefficient distribution of As and DVI spectral index.

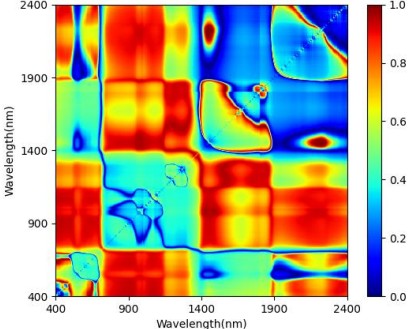

**Figure 7.** Correlation coefficient distribution of Pb and DVI spectral index.

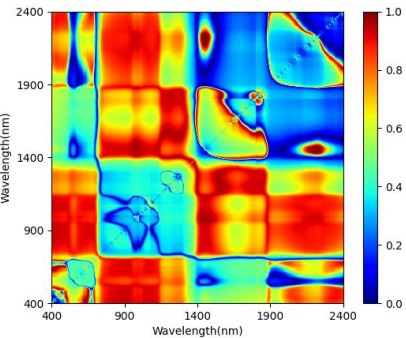

**Figure 8.** Correlation coefficient distribution of Cd and DVI spectral index.

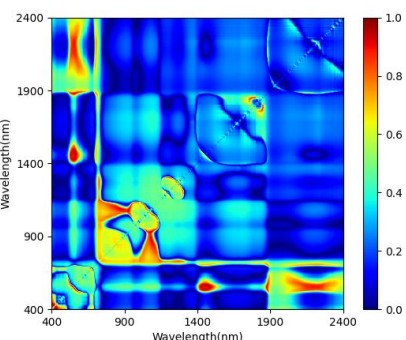

**Figure 9.** Correlation coefficient distribution of Cr and DVI spectral index.

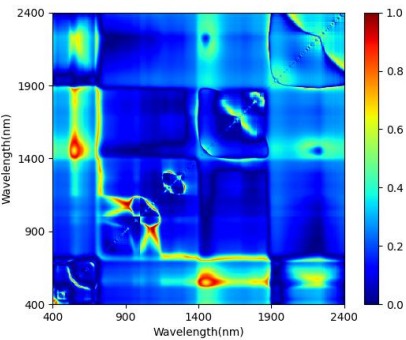

**Figure 10.** Correlation coefficient distribution of Hg and DVI spectral index.

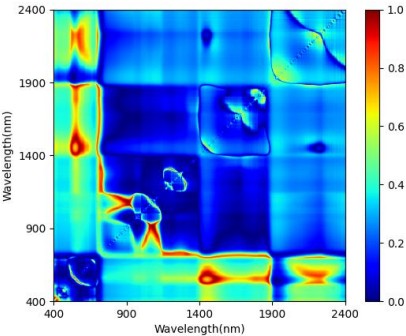

**Figure 11.** Correlation coefficient distribution of Cu and DVI spectral index.

In the distribution map, the coloring from blue to red indicates the correlation coefficients from small to large, and the deeper the color is, the greater the correlation coefficient is. Appendix A Figures A1–A3 show the correlation coefficient distribution diagrams constructed with the three elements As, Pb, and Cd and the four spectral indices, and their distributions are very similar. By comparison, the heavy metal content and DVI

have a wider sensitivity range, darker color, and larger correlation coefficient, indicating that the prediction model can be built more accurately using the difference spectral index of these three elements. In Figure A4, the correlation coefficient between Cr and DVI is relatively small, and the diagrams of Cr and SRVI, NDVI, and IDVI are similar; the correlation coefficients mostly fall between 0.5 and 0.7, and the correlation is medium. The distribution diagrams of the correlation coefficients of Hg and Cu and the four spectral indices in Figures A5 and A6 are generally blue, and the sensitive band range is narrow, which indicates that there are fewer band combinations with a strong correlation.

The wavelength positions where the heavy metal content was significantly correlated with the spectral index and the maximum correlation coefficient was located were extracted from Figures A1–A6, as shown in Table 4. It can be seen that the $r_{max}$ values of the spectral indices and heavy metal contents are all higher than 0.65, indicating that the correlation is good. Among them, the DVI has the highest correlation coefficient with each heavy metal. The band combination of the strongest correlation between the six heavy metals and the SRVI and NDSI was the same, and the correlation of NDSI was stronger than that of SRVI. The IDVI has the worst correlation with the different heavy metal elements. As, Pb, and Cd have the best correlation with DVI and NDSI at the 0.01 significance level; therefore, these three heavy metals were suitable for the prediction models of elemental content using the DVI and NDSI.

**Table 4.** Correlation coefficients between best band combinations of spectral index and heavy metals.

| Heavy Metal Element | Spectral Index | λ1 | λ2 | Correlation Coefficient |
|---|---|---|---|---|
| As | DVI | 655 | 480 | 0.711 ** |
| | SRVI | 605 | 603 | 0.705 * |
| | NDSI | 605 | 603 | 0.709 ** |
| | IDVI | 877 | 876 | 0.707 * |
| Pb | DVI | 651 | 451 | 0.711 ** |
| | SRVI | 604 | 603 | 0.702 * |
| | NDSI | 604 | 603 | 0.705 ** |
| | IDVI | 877 | 876 | 0.703 * |
| Cd | DVI | 646 | 432 | 0.711 ** |
| | SRVI | 604 | 603 | 0.706 * |
| | NDSI | 604 | 603 | 0.707 ** |
| | IDVI | 877 | 876 | 0.703 * |
| Cr | DVI | 572 | 528 | 0.699 * |
| | SRVI | 410 | 403 | 0.693 * |
| | NDSI | 410 | 403 | 0.695 * |
| | IDVI | 407 | 401 | 0.681 * |
| Hg | DVI | 914 | 786 | 0.687 * |
| | SRVI | 561 | 541 | 0.683 * |
| | NDSI | 561 | 541 | 0.684 * |
| | IDVI | 918 | 915 | 0.678 * |
| Cu | DVI | 596 | 520 | 0.700 * |
| | SRVI | 560 | 544 | 0.694 * |
| | NDSI | 560 | 544 | 0.695 * |
| | IDVI | 554 | 552 | 0.681 * |

Note: * Significant correlation at 0.05 level (double tail); ** significant correlation at 0.01 level (double tail).

### 3.5. Modeling and Accuracy Verification

#### 3.5.1. Prediction Model of Heavy Metal Elements Based on Single Variable

From the above study, it is clear that the DVI of specific band combinations has a high and significant correlation with the soil heavy metal elements. Therefore, in this paper, a regression analysis was conducted with the DVI as the independent variable and the soil heavy metal element content as the dependent variable to establish a prediction model of soil heavy metal element content in a univariate linear form.

The prediction model was fitted using training set data based on SPSS data analysis software. After the model was determined, the coefficient of determination ($R^2$) and

root mean square error (RMSE) were calculated using the test set to test the stability and prediction accuracy of the model. The coefficient of determination reflects the proportion of all the variations of the dependent variable that can be explained by the independent variable through the regression relationship. Generally speaking, a larger coefficient of determination indicates that the dependent variable can be better explained by the independent variable and the model has a better fit. Root mean square deviation is used to measure the deviation of the observed value from the true value, and the smaller the value is, the higher the stability of the model is. As shown in Table 5, the prediction model adapted with the DVI for As, Pb, and Cd content was correlated at a significance level of 0.05, and the order of the coefficient of determination was As > Pb > Cd. Of these, the DVI prediction model for As works best. The root mean square error (RMSE) was found to be large after testing, which indicates that the prediction model established using the difference spectral indices of the content of these three elements and their optimal band combinations are generally accurate and not stable. The DVI prediction models of Cr, Hg, and Cu heavy metals have a relatively poor fit, with poor prediction accuracy, and low stability.

**Table 5.** Prediction model of soil heavy metal elements.

| Heavy Metal | Spectral Index | Linear Model | $R^2$ | RMSE |
|:-----------:|:--------------:|:------------:|:-----:|:----:|
| As | R655-R480 | Y = −2772.418X + 59.774 | 0.612 * | 1.129 |
| Pb | R651-R451 | Y = −6368.798X + 174.448 | 0.606 * | 1.235 |
| Cd | R646-R432 | Y = −8.610X + 0.435 | 0.599 * | 1.298 |
| Cr | R572-R528 | Y = −5066.048X + 99.189 | 0.554 | 2.346 |
| Hg | R914-R786 | Y = 0.164X + 0.051 | 0.537 | 2.547 |
| Cu | R596-R520 | Y = 1039.642X + 24.706 | 0.566 | 3.336 |

Note: * Significant correlation at 0.05 level (double tail).

### 3.5.2. Prediction Model of Heavy Metal Elements Based on Multivariate

It can be seen from the results of the above research that the correlation is not significant between Cr, Hg, and Cu in soil and single band or spectral index. The elements As, Pb, and Cd are all sensitive to the blue, red, and long-wave near-infrared bands. At the same time, they have a certain significant correlation with the difference index and the normalized index. As a result, we identified three heavy metal elements suitable for fitting (As, Pb, and Cd) and five significantly correlated variables (blue band, red band, long-wave NIR, difference index, and normalized index). Due to the poor accuracy of the prediction models for soil heavy metal elements using a single spectral index, we used the five determined variables to perform multiple regression fittings with As, Pb, and Cd. The modeling can be constructed to express heavy metal content as a function or a set of functions and determine the parameters of the functions [38,39].

The multiple linear regression models were used to construct the prediction models for the content of soil heavy metal elements in SPSS software, and the results are presented in Table 6.

**Table 6.** Multiple regression model of soil heavy metal content.

| Heavy Metal | Multiple Regression Model |
|:-----------:|:-------------------------:|
| As | y = 40.655 + 18.587 × DVI − 178.328 × NDSI + 3.578 × Blue − 10.225 × Red + 11.6 × Near-infrared |
| Pb | y = 102.690 + 54.176 × DVI − 480.779 × NDSI + 16.729 × Blue − 44.332 × Red + 154.611 × Near-infrared |
| Cd | y = 0.267 + 0.073 × DVI − 0.547 × NDSI + 0.032 × Blue − 0.047 × Red − 0.116 × Near-infrared |

### 3.5.3. Prediction Model of Heavy Metal Elements Based on Multivariate

We use 20 samples from the test set and calculate the coefficient of determination ($R^2$) and root mean square error (RMSE) to test the stability and prediction accuracy of the model. The verification result is shown in Figure 12. The $R^2$ for As is 0.88786 and the

RMSE is 4.12329; the $R^2$ for Cd is 0.85641 and the RMSE is 0.04318; and the $R^2$ for Pb is 0.89387 and the RMSE is 0.88797. According to the results, we know that the inverse model of the soil heavy metal content has a good fit with the measured data, which shows that the model can forecast the heavy metal content of soil in the study area well.

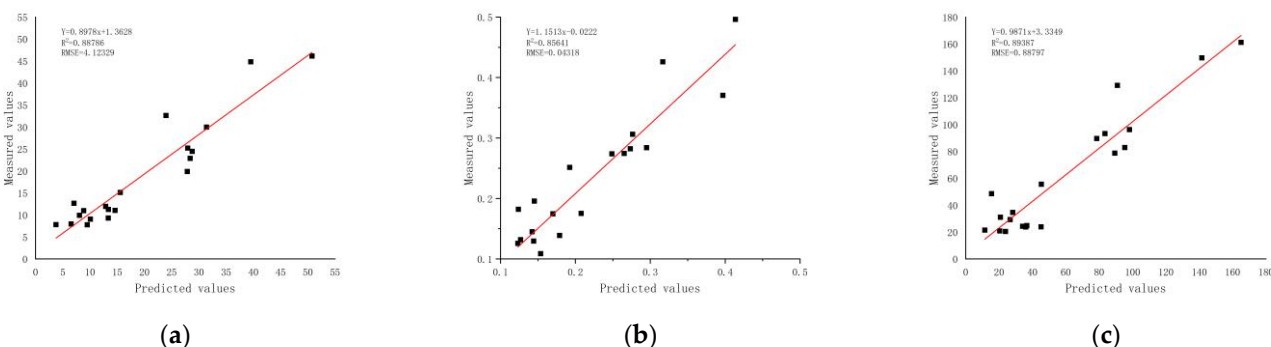

**Figure 12.** Accuracy verification of heavy metal content prediction model: (**a**) arsenic; (**b**) cadmium; (**c**) lead element.

## 4. Discussion

The peach tree is the most extensively planted fruit tree in the study area, and the production of the Pinggu big peach is famous in China and abroad, with the fruit being very popular among consumers. For this reason, we selected the peach tree as the subject of our study. We collected data from the same time period and the same climate in order to ensure data availability. By obtaining the mean value and the first-order derivative of the leaf spectral data, the effect of most of the background noise on the spectrum is eliminated, and the response characteristics of the original spectral curve can be accurately judged. Based on the environmental quality of the production area, the soil sampling points were reasonably set up in a representative manner and arranged to be collected at the late stage of crop growth. This series of data treatments excludes the main factors that affect the final results; therefore, the established model has remarkable precision and credibility.

The univariate linear regression equation is fast to model without a high complexity of calculation. It still works quickly with a large amount of data and can give an interpretation of each variable based on the coefficients. However, it cannot fit nonlinear data properly, which requires first determining whether the variables are linearly related to each other. There is more advantage of multiple regression models in nonlinear relationships. In fact, a phenomenon is often associated with multiple factors [40]. This is why it is more effective to predict or estimate the dependent variable by an optimal combination of multiple independent variables together than with only one independent variable, because it fits the reality of the situation more closely. We conducted a multiple regression analysis of the metals of interest with three spectral reflectance bands and two spectral indices. Compared with the ordinary linear regression model, this method improved the coefficients of determination for As, Pb, and Cd by 45.08, 41.32, and 49.23%, respectively, and reduced the root mean square error. Therefore, it is shown that the introduction of multiple variables with strong correlation to build a prediction model can provide a better fit to the measured data and validity for the prediction of soil heavy metal element content. Nowadays, machine learning methods are a popular technical approach and have good applications in this field [41]. However, most of the machine learning models are in a black-box state, and it is difficult for us to understand the internal mechanisms and, therefore, to explain the model results. Furthermore, some machine learning requires large amounts of data for training, which is difficult to maintain and requires high hardware configurations. On the contrary, it is relatively simple and rapid to build univariate linear regression models and multiple regression models. More importantly, it can explain the results easily, which makes it take an irreplaceable position in all aspects.

In this study, we collected a total of 58 sample point data and divided them into training and testing groups. Among them, 38 sample point data were used to construct

the prediction model of soil heavy metal elements, and the remaining 20 sample data were used for model validation. Overall, the selected sample size is relatively small. However, since the study area of this paper is not large, the sample size is statistically acceptable. We used indirect observations (vegetation spectra) to build the prediction model and the correlation of the calculated parameters was insufficient. The accuracy of the obtained models is not yet up to the standard of production practice, and future development is needed before they can be widely applied. Additionally, the enrichment of soil heavy metals to plants is a dynamic process that needs to take into account the influence of the physicochemical properties of the soil as well as biogeochemical barriers, etc. These factors have an impact on the entry of soil heavy metals into plants. In that case, the mechanism of heavy metal enrichment to plants and the consideration of the influencing factors into our current model are the future research development directions.

In the past, the investigation of soil heavy metals was often conducted with field plots [42–44], followed by chemical analysis indoors, and then geostatistical interpolation, which was expensive and time-consuming [45]. Compared with the traditional detection methods, the hyperspectral technique is relatively speedy and efficient, and it is a promising alternative method. Many authors used soil reflection spectra to establish the relationship with soil heavy metal elements in studies using hyperspectral techniques, while studies on the relationship between vegetation spectra and soil heavy metals are relatively lacking. Although such a method is a significant improvement over traditional chemical testing, it still requires access to the field for onsite measurements. There are some scholars who try to use the spectral response function to convert the narrow-wave spectral reflectance data collected from the ground into the wide-wave spectral reflectance data of remote sensing images, and then use the established inversion model to obtain the distribution of the soil heavy metal content in the study area, which attempts to construct a method to rapidly detect the intensity and distribution of regional soil heavy metal pollution. Such a method, however, obtains low accuracy and there are still many problems that need to be solved, such as the influence of vegetation cover, LAL, soil reflectivity, and other factors on satellite images. Nevertheless, the remote sensing images have the characteristics of periodicity, wide coverage, and high efficiency, which can avoid secondary pollution [46]. Highly accurate inversion models are constructed by vegetation spectral reflectance and applied to spatial inversion, which helps to monitor large areas quickly and effectively. The method possesses broad application prospects in the regulation of soil heavy metal pollution and could be applied to scientific research and production practice under certain precise conditions.

Certainly, there are differences in the climatic conditions, soil properties, and tree types in different geographical locations, and the degree of anthropogenic influence varies from region to region, both of which will affect the prediction accuracy of the model. Thus, we need to build the appropriate models for typical study areas under different conditions so that we can achieve sufficient accuracy. In the quantitative inversion of hyperspectral data, a direct relationship between elemental content and spectral features is usually found, and then a linear model is created as a way to predict the elemental content [47]. Such a method, however, has poor accuracy within the study area of this paper, while the regression model built with multiple variables has much better inversion capability. This offers reference value for other studies.

## 5. Conclusions

In this paper, in the field of the agricultural land environment, we analyzed the spectral characteristics of peach tree leaves under the stress of the soil heavy metal content with the dominant tree species in the study area. Further, the correlations between the different heavy metal elements and the spectral reflectance and spectral index of leaves were investigated. Finally, we constructed prediction models for As, Pb, and Cd contents and obtained good accuracy.

(1) Under different levels of heavy metal contamination, the trends of the reflectance spectral curves for the peach leaves were generally consistent, and the reflectance in the contaminated areas was generally higher than that in the background areas. There was a more obvious difference in the wavelength range of 760–1300 nm, which is relatively sensitive to the response of heavy metals in the soil. The heavy metals did not interfere significantly with the spectral positions of the red, blue, and yellow edges of the leaves, but they responded very obviously to the slopes of the three edges, which all increased with an increased heavy metal content.

(2) The correlations were weak and insignificant between the soil Cr, Cu, and Hg and spectral reflectance. The As, Pb, and Cd in the soil had strong correlations in some wavelength ranges, and the overall trends of the correlation curves were consistent. In comparison, the correlation between the soil heavy metal elements and the differential spectral indices was greater, with correlation coefficients all greater than 0.65. Subsequently, we used SPSS data analysis software to establish a one-dimensional linear regression model of different heavy metal elements and indices, and it was found that the accuracy of the model was moderately accurate and not quite stable.

(3) The study used five variables (blue band, red band, long-wave NIR, difference index, and normalized index) as independent variables and selected the soil heavy metal elements (As, Pb, and Cd) with a stronger correlation and significance to establish the multiple linear regression model. We randomly selected 20 sample points to test the prediction model for each of the three elements, and we evaluated the accuracy of the model inversion using the $R^2$ and RMSE. In particular, the inversion accuracy of Pb was the highest and Cd was the lowest. The overall reproduction accuracy ($R^2$) of the model is greater than 0.8 and the RMSE is less than 5, which indicates that the predictive inversion model can effectively estimate the contents of As, Pb, and Cd.

**Author Contributions:** W.L. conceived the idea, performed data treatments, and wrote and revised the paper; Q.Y. contributed some ideas and revised the paper; T.N., L.Y., and H.L. contributed to the discussion of the results. All authors have read and agreed to the published version of the manuscript.

**Funding:** This research was funded by the National Natural Science Foundation of China Youth Science Foundation Project (42001211).

**Data Availability Statement:** The leaf spectral data and the soil heavy metal content in this study.

**Conflicts of Interest:** The authors declare no conflict of interest.

## Appendix A

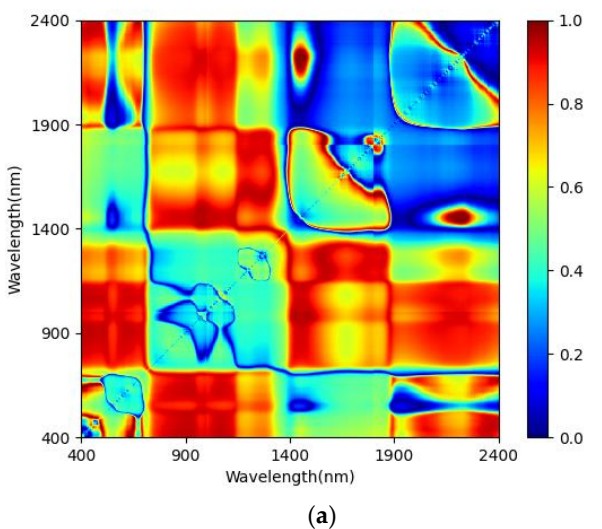
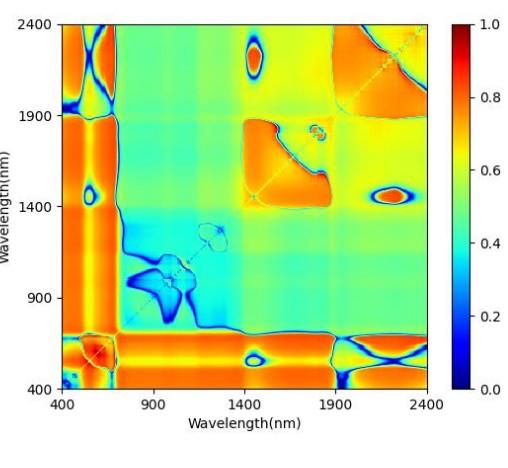

(a)                              (b)

**Figure A1.** *Cont.*

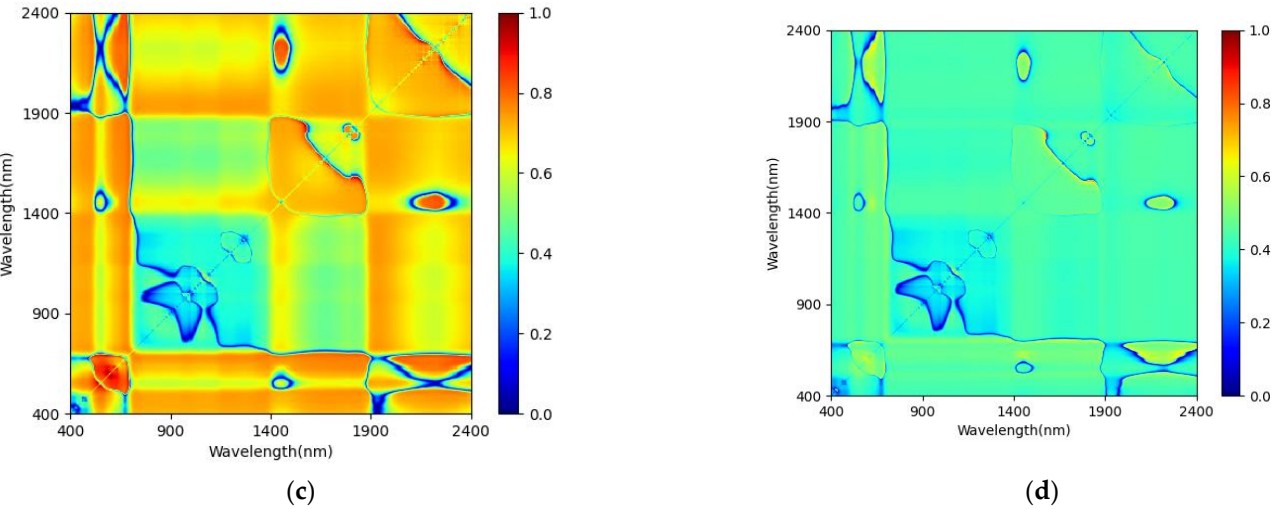

**Figure A1.** Correlation distribution of As content and spectral index: (**a**) DVI, (**b**) SRVI, (**c**) NDVI, (**d**) IDVI.

**Figure A2.** Correlation distribution of Pb content and spectral index: (**a**) DVI, (**b**) SRVI, (**c**) NDVI, (**d**) IDVI.

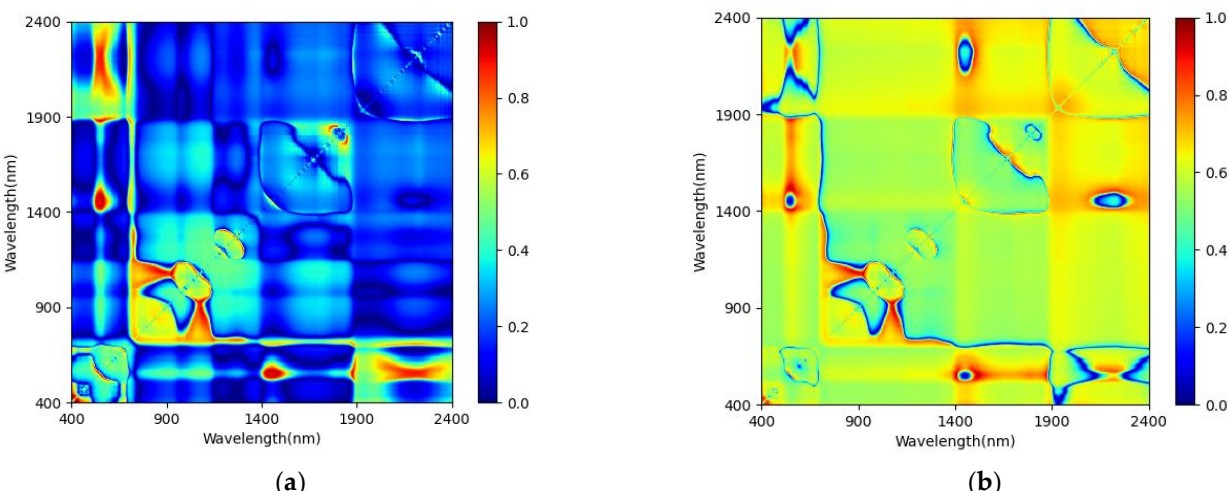

**Figure A3.** Correlation distribution of Cd content and spectral index: (**a**) DVI, (**b**) SRVI, (**c**) NDVI, (**d**) IDVI.

(**a**)

(**b**)

**Figure A4.** *Cont.*

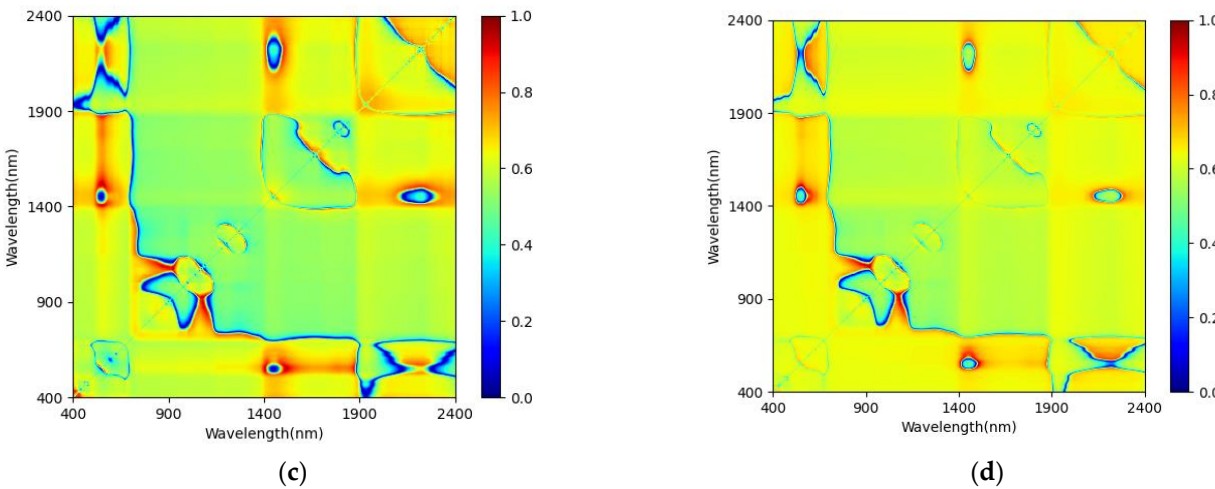

**Figure A4.** Correlation distribution of Cr content and spectral index: (**a**) DVI, (**b**) SRVI, (**c**) NDVI, (**d**) IDVI.

**Figure A5.** Correlation distribution of Hg content and spectral index: (**a**) DVI, (**b**) SRVI, (**c**) NDVI, (**d**) IDVI.

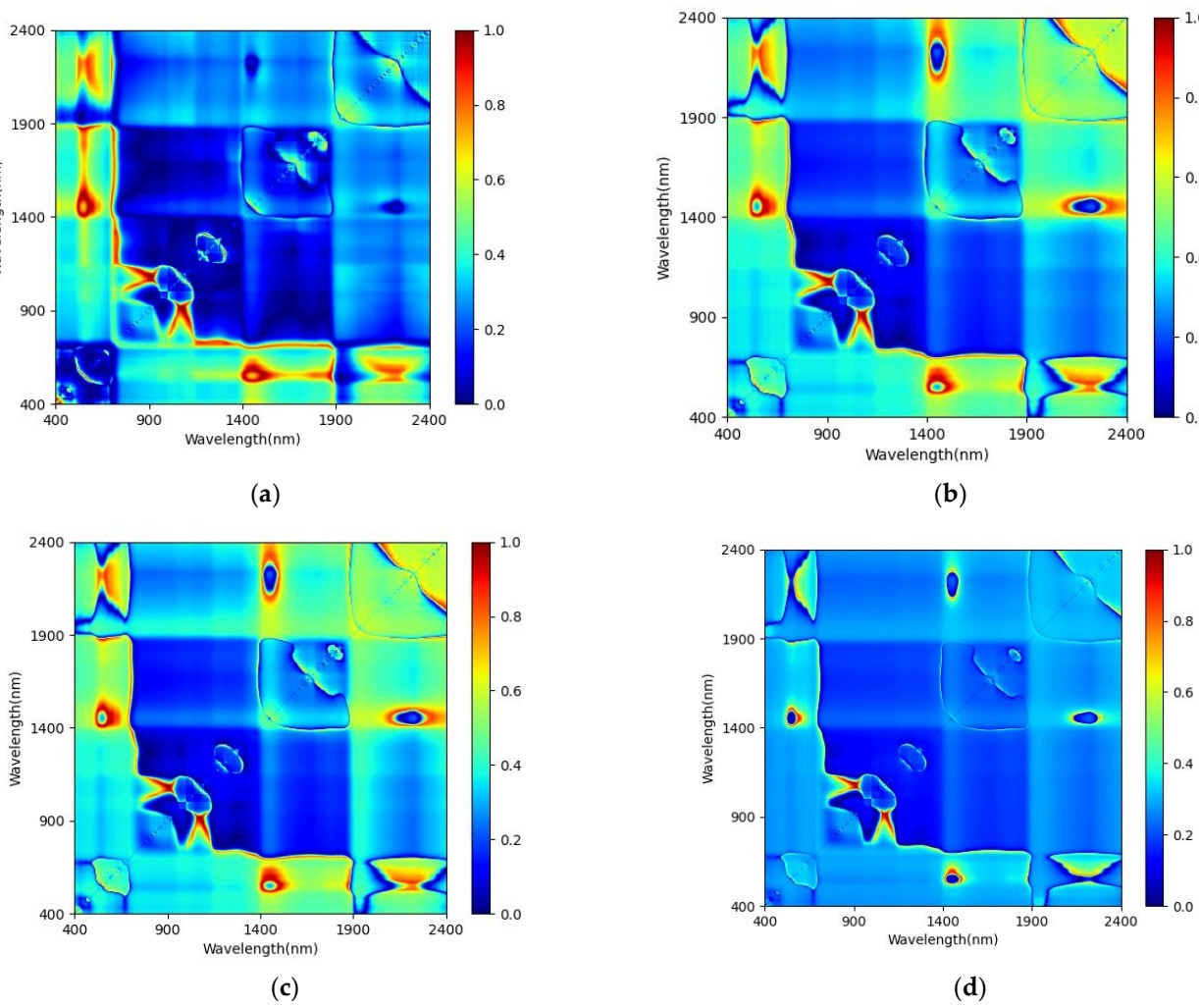

**Figure A6.** Correlation distribution of Cu content and spectral index: (**a**) DVI, (**b**) SRVI, (**c**) NDVI, (**d**) IDVI.

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
