# Peer review of "Inversion of Soil Heavy Metal Content Based on Spectral Characteristics of Peach Trees"

_forests, doi:10.3390/f12091208_

Round 1

Reviewer 1 Report

Dear Authors,

Your contribution is important, but needs for revision. There are some minor flaws. Some statements are to be clarified.

There are many examples of the hyperspectral methodology applications to study the soil heavy metal contamination in China (https://doi.org/10.1016/j.catena.2021.105222).

What is a fundamental difference between your methodology and the previously used well-known approaches? What is a novelty of your method for predicting soil heavy metal content by hyperspectral techniques?

Line 43

ecological environment

“ecological” is a redundant word here.

Line 55

19].At

Provide the space before “At”.

Line 56

domestic and foreign scholars

Why should we divide the scholars? Science is an international (maybe even over-national) field of the Human’s intellectual accomplishments. Moreover, the term “domestic” is occupied in another field of science. This term is consistently associated with cattle breeding, separating domestic and wild animals. My advice to you: use “Chines scholars and foreign scholars” or simply “scholars”.

Line 73

NDVI705

NDVI (Normalized Difference Vegetation Index) is a widespread term. At the same time, I think, the rule to disclose the abbreviation at first use is not bad.

Line 99

Landset8

Pease correct the spelling.

Line 105

70km

Pease provide the space.

Line 108

Data on the precipitation and temperature regime are desirable. Maybe some more characteristics of the region.

Line 115

product. And

Delete the full stop sign.

Line 117

there are nine tailings ponds

The location of the source of pollution needed to understand the pollutant spread. This is important (maybe – main) condition for the proper assessment of environment. You did not presented the localization of the ponds as the sources of pollution of the study area. This does not provide a reliable sampling strategy and reliability of your data and model as well.

Lines 134-137

We adopted a combination of environmental units and administrative units, that is, based on comprehensive consideration of geographical topography, soil texture and other environmental factors, we rationally set up 58 sampling points with administrative village agricultural land as a unit (Figure 1).

The sampling strategy you presented is unconvincing.

Line 155
40cm

Space?

Line 163

60°

60ᵒC (?)

Line 191

2500nm

Space?

Lines 193, 194

Four spectral index calculation formula as shown in table 1.

The phrase seems awkward.

Line 196

2500nm

Space?

Line 222

cinnamon soil

Please characterize the soil in the Methods.

Line 231

Table 2.

The heavy metal content values are low. Mean data are lower than National Grade II Standard. Environment pollution becomes noticeable and significant at the higher level of heavy metal content. In the range the authors presented, the biological regulation (biogeochemical barriers) provide control of the heavy metal content in the plant.

Line 240

the background value

How did you find this value?

Line 259

760nm

Space?

Line 276

[31]

Superscript?

Lines 306, 307

The correlation coefficient between soil Cu element and reflectance is between 0.14 and 0.42, and some wavebands have weak correlation.

The word “and” indicates for the readers that a correlation of 0.42 is, allegedly, sufficient as a sign of the relationship. In fact, this correlation level is small and is not be discussed as an indicator of a significant relationship between the parameters.

Line 316

515 – 674

You have used the different signs for the range. Please accept the only one.

Line 376

Table 4

You use two grades of the correlation significance in vain. The maximum level 0.711** and the minimum level 0.678* are the same. In statistics, the well-known criterion of the correlation significance is a coefficient of determination. D=R2≥0.5. In your case, we have 0.505 and 0.461 – both D value are the same. 

Lines 506-510

At the same time, the accuracy is required higher for such a method that models indirectly through vegetation spectra. As for the complex soil environment, we will conduct further deeper research combined with other prediction methods, such as deep learning, support vector machine and random forest.

This phrase in the end of the Discussion is modest and real. Indeed, you used indirect observation results. The level of correlation of the calculated parameters is not high enough. In addition, biogeochemical barriers, which you did not take into account, have a decisive effect on the migration and access to the plants of heavy metals due to the soil moisture regime, calcium-carbonate equilibrium in the soil solution, organic matter quantity and dynamics, and many other soil equilibria. When the soil equilibrium changes, the picture you have presented may change too, and fundamentally.

We recommend you revise the Conclusions slightly in the vein of humble self-esteem. The Conclusions in the current form differ from the final phrase of the Discussion for the worse. As it seems to us, the volume of Conclusions is excessive, and the results obtained are overestimated.

Author Response

1、There are many examples of the hyperspectral methodology applications to study the soil heavy metal contamination in China (https://doi.org/10.1016/j.catena.2021.105222).

What is a fundamental difference between your methodology and the previously used well-known approaches? What is a novelty of your method for predicting soil heavy metal content by hyperspectral techniques?

            First of all, I would like to thank the experts for recommending the references. This paper uses a more direct approach: soil reflectance spectroscopy for soil heavy metal prediction, which also makes the results more accurate and reliable, and the support vector machine and random forest model used are very inspiring.

The most fundamental difference compared to previous well-known methods is that this paper is not limited to the combined bands of spectral indices studied by previous authors, but all combined bands of typical spectral indices are calculated. Unlike the direct use of soil reflectance spectra, the indirect use of vegetation reflectance spectra for the inversion of soil heavy metal content was adopted due to the consideration of vegetation cover, which is the novelty of this paper.

2、Line 43

ecological environment

“ecological” is a redundant word here.

            The word “ecological” has been deleted from the original text. The original text was amended to read: “which makes them a major safety hazard to human health and the environment.”

3、Line 55

[19]. At

Provide the space before “At”.

A space has been added before "At" in the original text, and a separate line has been added to divide it into two paragraphs to make the text clearer.

4、Line 56

domestic and foreign scholars

Why should we divide the scholars? Science is an international (maybe even over-national) field of the Human’s intellectual accomplishments. Moreover, the term “domestic” is occupied in another field of science. This term is consistently associated with cattle breeding, separating domestic and wild animals. My advice to you: use “Chinese scholars and foreign scholars” or simply “scholars”.

Thank you very much for the guidance from the expert teachers. The original text has been changed to "scholars".

5、Line 73

NDVI705

NDVI (Normalized Difference Vegetation Index) is a widespread term. At the same time, I think, the rule to disclose the abbreviation at first use is not bad.

Thanks to the guidance of the teacher, the term NDVI has been explained in the first use.

6、Line 99

Landset8

Pease correct the spelling.

The word has been revised.

7、Line 105

70km

Pease provide the space.

The original formatting has been revised to add spaces.

8、Line 108

Data on the precipitation and temperature regime are desirable. Maybe some more characteristics of the region.

I appreciate the comments given by the expert teachers. Precipitation and temperature conditions in the area have been added. The original article was inserted: “The annual average temperature is 11.7 °C, the coldest in January with an average temperature of -5.4 °C and the hottest in July with an average temperature of 26.1 °C. The annual precipitation is 629.4 mm, mainly concentrated in summer, at 453.0 mm, which is 72% of the annual precipitation.”

9、Line 115

product. And

Delete the full stop sign.

The English expression in the original text has been modified to conform to the specification. The original text was modified to read: “Peach trees are widely planted in this area, and the Pinggu big peach is famous in the market and is a Chinese geographical indication product. The area contains a variety of minerals such as gold, silver, copper, and manganese.”

10、Line 117

there are nine tailings ponds

The location of the source of pollution needed to understand the pollutant spread. This is important (maybe – main) condition for the proper assessment of environment. You did not presented the localization of the ponds as the sources of pollution of the study area. This does not provide a reliable sampling strategy and reliability of your data and model as well.

We are grateful to the expert teachers for your comments. The "Figure 1. Study area and distribution of sampling points." has been redrawn with the geographic location of the tailings ponds in the study area.

The original text is incorrectly written and has been revised. The location of the study area is Pinggu District, Beijing, China, and the original text should have stated that there are nine tailing ponds in Pinggu District, Beijing. The study area was selected as the more polluted Yukou area in Pinggu District, and there are two tailing ponds, which have been marked in Figure 1.

11、Lines 134-137

We adopted a combination of environmental units and administrative units, that is, based on comprehensive consideration of geographical topography, soil texture and other environmental factors, we rationally set up 58 sampling points with administrative village agricultural land as a unit (Figure 1).

The sampling strategy you presented is unconvincing.

The original description of the sampling strategy has been refined and expressed in detail to make it more convincing. The original text was amended to read: “Along the diagonal position of the square sample plot, at the two top corners as well as at the geometric center, we respectively selected a mature peach tree 5-8 years old, which satisfied a height of 5-6 m and a diameter at breast height of 8-10 cm. At the same time, it records in detail the geographical coordinates, ambient temperature and humidity, solar radiation, etc. After the peach trees were identified, we could start measuring the leaf spectra by selecting 5-10 leaves per tree and collecting 10 spectral data points per leaf. We selected mature dark green peach leaves and required intact, healthy and disease-free leaves.”

12、Line 155

40cm

Space?

The original formatting has been revised.

13、Line 163

60° 

60ᵒC (?)

Revised the original text to “60ᵒC”

14、Line 191

2500nm

Space?

The original formatting has been revised.

15、Lines 193, 194

Four spectral index calculation formula as shown in table 1.

The phrase seems awkward.

Modify the phrase to “The four spectral indices were calculated as shown in Table 1.”

16、Line 196

2500nm

Space?

The original formatting has been revised.

17、Line 222

cinnamon soil

Please characterize the soil in the Methods.

The original description of the soil has been placed under "Methods".

18、Line 231

Table 2.

The heavy metal content values are low. Mean data are lower than National Grade II Standard. Environment pollution becomes noticeable and significant at the higher level of heavy metal content. In the range the authors presented, the biological regulation (biogeochemical barriers) provide control of the heavy metal content in the plant.

We reworked this paragraph and added coefficients of variation to the table. Although the average values of heavy metals in the study area do not achieve the national secondary standards, the overall situation is relatively optimistic. However, the maximum value is far beyond the secondary standard data, so the study is also necessary. We are grateful for your comments and have corrected the original expression to better represent the significance of the study.

The original text was amended to read: “It can be seen from the table that the coefficient of variation was moderate between 10% and 100%. The mean values of all heavy metals exceeded the background values of the soil environment in the 58 samples collected. The average values of three metals, Cr, As and Pb, are higher than the national grade I standard, and the rest are lower than the standard value. Besides, all the metals were lower than the national grade II standard, showing that the soil environmental quality was generally passable in the area. However, from the maximum value, the content of six heavy metal elements exceeded the national level standard, and some heavy metals exceeded the secondary standard, which indicated that the soil environment in the study area was polluted within different degrees, and had a certain impact on the cultivation of peach trees.”

19、Line 240

the background value

How did you find this value?

The background values are quoted from the book “Monitoring and Evaluation of Environmental Quality of Agricultural Produce”, and have been cited in the original text.

20、Line 259

760nm

Space?

The original formatting has been revised.

21、Line 276

[31]

Superscript?

The original formatting has been revised.

22、Lines 306, 307

The correlation coefficient between soil Cu element and reflectance is between 0.14 and 0.42, and some wavebands have weak correlation.

The word “and” indicates for the readers that a correlation of 0.42 is, allegedly, sufficient as a sign of the relationship. In fact, this correlation level is small and is not be discussed as an indicator of a significant relationship between the parameters.

The original text has been amended to read: “The correlation coefficients of soil Cu and reflectance fluctuated in the range of 0.14–0.42, which revealed that there was a weak linear correlation between Cu and spectral reflectance.”

It is considered that the use of correlation coefficient r can be used as a criterion for evaluating the linear correlation of two variables, hence the use of correlation coefficient in the article. The original article was modified to specify that the correlation here is a linear correlation and that the correlation is weak. The correlation between the six heavy metal elements and the spectral reflectance in the article is not all very significant. However, considering the soil statistics in the previous subsection 3.1, certain Cu elements are heavily contaminated, so it is considered that there is still some research value. And it can be seen that Cu can be correlated up to 0.42 in some wavelength range, indicating that it still has a high research value in this range.

23、Line 316

515 – 674

You have used the different signs for the range. Please accept the only one.

The original formatting has been revised.

24、Line 376

Table 4

You use two grades of the correlation significance in vain. The maximum level 0.711** and the minimum level 0.678* are the same. In statistics, the well-known criterion of the correlation significance is a coefficient of determination. D=R2≥0.5. In your case, we have 0.505 and 0.461 – both D value are the same. 

Taking into account that the article is required to perform a linear correlation fit first, and that the correlation coefficient r is exactly able to evaluate the linear correlation between the two variables, hence R2 was not used as an indicator for the calculation.

25、Lines 506-510

At the same time, the accuracy is required higher for such a method that models indirectly through vegetation spectra. As for the complex soil environment, we will conduct further deeper research combined with other prediction methods, such as deep learning, support vector machine and random forest.

This phrase in the end of the Discussion is modest and real. Indeed, you used indirect observation results. The level of correlation of the calculated parameters is not high enough. In addition, biogeochemical barriers, which you did not take into account, have a decisive effect on the migration and access to the plants of heavy metals due to the soil moisture regime, calcium-carbonate equilibrium in the soil solution, organic matter quantity and dynamics, and many other soil equilibria. When the soil equilibrium changes, the picture you have presented may change too, and fundamentally.

We recommend you revise the Conclusions slightly in the vein of humble self-esteem. The Conclusions in the current form differ from the final phrase of the Discussion for the worse. As it seems to us, the volume of Conclusions is excessive, and the results obtained are overestimated.

We are grateful to the expert teachers for their guidance and comments, which are important for our research. The issues mentioned by the teachers have been added to the discussion section of the original paper, which is also the direction of our future research.

Reviewer 2 Report

This articles aims at estimating the heavy metal contents in leaf samples using reflectance data by comparing various regression models. This research is interesting and brings new evidence of the feasibility to retrieve leaf metals in leaves using field spectroscopy. However, several points must be addressed before it can be published.

Introduction:

- L47 “this has become a hot subject of research for scholars”. This sentence is not appropriate. Please rephrase it in a more scientific way.

- L50-54 I strongly disagree with these assumptions. Optical remote sensing (plane, satellite, and reflectance spectroscopy) requires clear, cloud-free weather since it is a passive technique. Also, please provide references to support the following assumption: “realize large-scale and quantitative extraction of soil heavy metal content”.

- L55 “many domestic and foreign scholars”. What do you mean by “domestic” and “foreign” here? Do you mean researchers from China, and foreign researchers? There is no need to distinguish the nationality of research teams in a scientific article!

- L58 “they have made many breakthroughs”. This is too vague. Please rephrase this sentence or remove it.

- L73-76 Heavy metal estimation from soil reflectance is called “direct” estimation, whereas heavy metal estimation from vegetation reflectance is called “indirect” estimation. Please use the appropriate vocabulary.

- L92-100 This paragraph is a synthesis of the abstract. Please remove it.

- From a general point of view, the introduction is lacking of discussion on the limits of the current methods of heavy metal estimation using reflectance spectroscopy. Many examples from past studies are presented in a vague way. After reading the introduction, we should understand what are the current limits/issues in that field and how the proposed approaches will overcome them. Likewise, there is no discussion on scaling up the approaches, from laboratory to imagery scales.

Materials and methods:

This section is poorly written and is lacking of detailed protocol. Please provide more details on sampling, measurement conditions, image processing, in order to make your work reproducible.

- L 133-135 This is too vague! Details are needed here. How did you select the peach trees, which leaves did you sample (young, mature, etc.)? Did you measure environmental variables (what is the “geographic environment” you mention)?

- L 136 Please avoid the term “reflection”, and use “reflectance” instead. Also, please note that the use of “→” is NOT appropriate in a scientific article.

- L144 What do you mean by “abnormal” spectral curve?

- L 144-145 This is not an appropriate sentence. Here, you give a definition, instead of explaining that you averaged the spectra.

- My main concern with paragraph 2.2.1 is that many details are missing. What is the angle of the sensor (nadir?), and the zenith angle during the measurements? Did you use a fore optics and place the sensor at a given distance from the sample? If so, what is the resulting acquisition footprint? Did you measure radiance and used a white calibration Spectralon to retrieve reflectance? If so, please mention it.

- Figure 2 “Sensor” instead of “Senser” on the scheme.

- L 149-150 Did you sample the soil in the 0-40 cm and mix it together?

- L 159-165 Details on the laboratory analysis carried out to estimate soil heavy metals are missing here. Soil analysis is a critical step of your research since it served as a reference for assessing your reflectance models. So, please specify the type of analysis performed, the entire protocol, the reagents used, etc.

- Paragraph 2.2.3 Why did you use Landsat 8 imagery? I would consider using Sentinel-2 imagery as it provides 10-20 m spatial resolution with up to 16 spectral bands! There are Sentinel-2 images available over your study area.

- L 174-177 Please give more details about the pre-processing of the image. What algorithms did you use to perform radiometric calibration, atmospheric compensation, noise reduction, etc.? Please also provide reference for each pre-processing step, as well as the date of image acquisition.

- L 185 You computed the spectral indices over the [350-2500] spectral domain. However, the [350:400] and [2400:2500] nm regions are extremely noisy. As recommended in many past studies, I would consider using only the [400:2400] nm domain.

- L 192-206 This paragraph is poorly written and must be rephrased. How did you apply the regression models? Did you split the dataset in training and test sets? If so, how did you select the training and test samples? Please refer to the following studies for guidelines about how to train, apply, and evaluate regression models using reflectance spectroscopy:

Lassalle, G., Fabre, S., Credoz, A., Hédacq, R., Bertoni, G., Dubucq, D., & Elger, A. (2019). Application of PROSPECT for estimating total petroleum hydrocarbons in contaminated soils from leaf optical properties. Journal of Hazardous Materials, 377(February), 409–417. https://doi.org/10.1016/j.jhazmat.2019.05.093

Shi, T., Liu, H., Chen, Y., Wang, J., & Wu, G. (2016). Estimation of arsenic in agricultural soils using hyperspectral vegetation indices of rice. Journal of Hazardous Materials, 308, 243–252. https://doi.org/10.1016/j.jhazmat.2016.01.022

- L 192-206 You mention using the correlation coefficient (r, not R). What statistical test did you apply to verify the correlation? Spearman, Pearson, Kendall correlation test?

- L 192-206 There is no mention of the metrics used to assess the model. Did you compute the R², the RMSE, the %RMSE, the RPD? These metrics are highly relevant and MUST be defined here (principle, formula, interpretation, references).

Results:

-- All the graphical figures (not maps) from this section are unreadable. They MUST appear with the appropriate font size, line color and thickness, figure size and resolution. The manuscript cannot be published with the current figure quality. Important improvements are expected here.

- Table 2 Please add the coefficient of variation for each metal here(CV in %).

- Figure 3 How did you compute the “difference” spectrum. It seems that you subtracted the average contaminated spectrum and the background one. However, there is no mention of that in the Materials and Methods section. Also, I recommend adding the variability (standard-deviation or min-max) for the two average spectra here, in order to better interpret the spectral differences.

- Figure 5 Were the correlations computed using the derivative spectra? If so, please mention it in the figure caption. Also, several heavy metal show the same correlation curve here. How are these metals correlated with each other? For example, if soil Hg and Pb are correlated, there is no need to develop a model for each metal.

- Figure 6-11 are unreadable and most of these figures are irrelevant. I would recommend showing only the “best” heatmap for each metal, and placing the other figures as appendices.

- Table 4 Please modify this table so that each metal appear row after row, not as columns (Cr, Hg, and Cu must appear below Cd).

- The results presented in section 3.5 must be re-computed using the approach mentioned in a previous comment (train + test and appropriate metrics). These results cannot be published if they are not produced in an appropriate way.

- Paragraph 3.5.2 This paragraph belongs to the Materials and Methods section.

- Paragraph 3.6 This component of the research (mapping) is not appropriate for publication. It is of poor scientific quality and I cannot recommend the article for publication if this “mapping” of heavy metals remains. You apply a regression model developed at leaf scale to 30-m Landsat images with no intermediary step (for example, canopy measurements). Satellite reflectance data are way different than leaf reflectance data. Canopy reflectance as viewed by satellite is influenced by canopy cover density, LAI, soil reflectance, leaf orientation, and many other parameters. Applying a leaf-based model directly to satellite imagery is pure science-fiction. Moreover, you pretend to assess the model’s accuracy on Landsat imagery (L 432-439), but you are linking soil sampled over a few centimeter area to 30-m pixels! I think you should limit you research to leaf-scale measurements here, and remove this non-scientific extension of your research.

Discussion:

- This section is poorly written. It is a simple synthesis of the results obtained by the proposed models. There is not comparison with previous work, no discussion about the limitations of the study. This section is too short, too vague, and written in a non-scientific way (the conclusion is longer than the discussion!).

Author Response

L47 “this has become a hot subject of research for scholars”. This sentence is not appropriate. Please rephrase it in a more scientific way.

Thank you for the expert teacher's correction. The original text has been amended to read: “and many scholars have been trying to monitor the soil environment using hyperspectral techniques.”

2、- L50-54 I strongly disagree with these assumptions. Optical remote sensing (plane, satellite, and reflectance spectroscopy) requires clear, cloud-free weather since it is a passive technique. Also, please provide references to support the following assumption: “realize large-scale and quantitative extraction of soil heavy metal content”.

However, hyperspectral remote sensing technology can be free from the restrictions of terrain and climate, break through the vegetation barrier, realize large-scale and quan-titative extraction of soil heavy metal content, and meet the application requirements of timeliness and accuracy, so it has gradually developed and matured

Thank you for the expert teacher's correction. The original text has been amended to read: “In contrast, hyperspectral remote sensing technology is quicker and more efficient, has high resolution and accuracy, which makes it more suitable for monitoring at large spatial scales. It is a promising alternative detection technology and is gradually developing and maturing.”

3、- L55 “many domestic and foreign scholars”. What do you mean by “domestic” and “foreign” here? Do you mean researchers from China, and foreign researchers? There is no need to distinguish the nationality of research teams in a scientific article!

Thank you for the expert teacher's correction. The original text has been amended to read: “At present, many scholars have conducted research on heavy metal pollution in soil by using ground object reflectance spectroscopy.”

4、- L58 “they have made many breakthroughs”. This is too vague. Please rephrase this sentence or remove it.

The original text has been removed here.

5、- L73-76 Heavy metal estimation from soil reflectance is called “direct” estimation, whereas heavy metal estimation from vegetation reflectance is called “indirect” estimation. Please use the appropriate vocabulary.

The original text has been changed to read : “direct estimation of heavy metal content by soil reflectance spectroscopy and indirect estimation of heavy metal content by vegetation reflectance spectroscopy.”

6、- L92-100 This paragraph is a synthesis of the abstract. Please remove it.

The original text has been removed here.

7、- From a general point of view, the introduction is lacking of discussion on the limits of the current methods of heavy metal estimation using reflectance spectroscopy. Many examples from past studies are presented in a vague way. After reading the introduction, we should understand what are the current limits/issues in that field and how the proposed approaches will overcome them. Likewise, there is no discussion on scaling up the approaches, from laboratory to imagery scales.

The introduction section of the original text has been revised to add the shortcomings of past research examples, point out the current problems and limitations in the field, and the necessity of building appropriate models for different regions. Also, the comparison of modeling in laboratory and field environments and the problems faced have been added to the original text.

8、- L 133-135 This is too vague! Details are needed here. How did you select the peach trees, which leaves did you sample (young, mature, etc.)? Did you measure environmental variables (what is the “geographic environment” you mention)?

Thank you for the correction. We have revised the original text to give more detail on the conditions for selecting peach trees and leaves, and the specific environmental variables (temperature and humidity, solar radiation, etc.)

9、- L 136 Please avoid the term “reflection”, and use “reflectance” instead. Also, please note that the use of “→” is NOT appropriate in a scientific article.

Thank you for the reminder, and the original text has been revised.

The original text was rewritten where the symbol "→" was included. The original text was amended to read: “Step 1 optimize the spectrometer, followed by scanning the standard white plate for calibration, and start measuring when it collects a straight horizontal footprint for a certain period of time; Step 2 adjust the projection mode, then make sure the probe is 5-10 cm vertically down from the leaf surface; Step 3 utilize the instrument to collect data, and take care to keep the value stable after reading.”

10、- L144 What do you mean by “abnormal” spectral curve?

The abnormal spectral curve refers to the fact that the overall reflectance of the curve is significantly higher or lower than the normal vegetation reflectance spectrum, and these may be caused by too much sunlight or lack of sunlight, so they are deleted.

11、- L 144-145 This is not an appropriate sentence. Here, you give a definition, instead of explaining that you averaged the spectra.

The original text has been revised here. The original text was amended to read: “Finally, the average value of all data is calculated as the actual reflectance information of this point.”

12、- My main concern with paragraph 2.2.1 is that many details are missing. What is the angle of the sensor (nadir?), and the zenith angle during the measurements? Did you use a fore optics and place the sensor at a given distance from the sample? If so, what is the resulting acquisition footprint? Did you measure radiance and used a white calibration Spectralon to retrieve reflectance? If so, please mention it.

We have revised most of subsection 2.2.1 and added the details mentioned by the teacher, thanks for your comments.

13、- Figure 2 “Sensor” instead of “Senser” on the scheme.

The images in the article have been replaced.

14、- L 149-150 Did you sample the soil in the 0-40 cm and mix it together?

Yes. A sampling site contains multiple sub-sampling sites, and the soil from all sub-sampling sites is mixed well before it is used as the final sample for testing.

15、- L 159-165 Details on the laboratory analysis carried out to estimate soil heavy metals are missing here. Soil analysis is a critical step of your research since it served as a reference for assessing your reflectance models. So, please specify the type of analysis performed, the entire protocol, the reagents used, etc.

The original text has been supplemented with a section on laboratory analysis of heavy metal elements in soil, adding details on the types of analysis, reagents, and methods. Modify the original text to read: “The soil PH value is determined by using the spot method through a PH meter. The soil organic matter content was tested using the potassium dichromate volumetric method through an electric furnace. The element cadmium (Cd) was measured using graphite furnace atomic absorption spectrophotometry by atomic absorption spectrometer. The chromium (Cr), lead (Pb) and copper (Cu) were analyzed using flame atomic absorption spectrophotometry by atomic absorption spectrometer; The amount of total mercury (Hg) was obtained using the cold atomic absorption method by a mercury measuring instrument; The arsenic (As) content was quantified using silver diethyldithiocarbamate spectrophotometric method by spectrophotometer.”

16、- Paragraph 2.2.3 Why did you use Landsat 8 imagery? I would consider using Sentinel-2 imagery as it provides 10-20 m spatial resolution with up to 16 spectral bands! There are Sentinel-2 images available over your study area.

After careful consideration of the comments made by the teacher, it was decided to remove the part of the spatial inversion that was not of high scientific quality.

17、- L 174-177 Please give more details about the pre-processing of the image. What algorithms did you use to perform radiometric calibration, atmospheric compensation, noise reduction, etc.? Please also provide reference for each pre-processing step, as well as the date of image acquisition.

After careful consideration of the comments made by the teacher, it was decided to remove the part of the spatial inversion that was not of high scientific quality.

18、- L 185 You computed the spectral indices over the [350-2500] spectral domain. However, the [350:400] and [2400:2500] nm regions are extremely noisy. As recommended in many past studies, I would consider using only the [400:2400] nm domain.

Thank you for your comments. The original chart and data have been recalculated to use only the less noisy 400-2400 nm band range.

19、- L 192-206 This paragraph is poorly written and must be rephrased. How did you apply the regression models? Did you split the dataset in training and test sets? If so, how did you select the training and test samples? Please refer to the following studies for guidelines about how to train, apply, and evaluate regression models using reflectance spectroscopy:

Lassalle, G., Fabre, S., Credoz, A., Hédacq, R., Bertoni, G., Dubucq, D., & Elger, A. (2019). Application of PROSPECT for estimating total petroleum hydrocarbons in contaminated soils from leaf optical properties. Journal of Hazardous Materials377(February), 409–417. https://doi.org/10.1016/j.jhazmat.2019.05.093

Shi, T., Liu, H., Chen, Y., Wang, J., & Wu, G. (2016). Estimation of arsenic in agricultural soils using hyperspectral vegetation indices of rice. Journal of Hazardous Materials308, 243–252. https://doi.org/10.1016/j.jhazmat.2016.01.022

Thanks to the article recommended by the teacher, this paragraph has been rewritten. The article details the formula algorithm for the regression model, how to group the data set and the accuracy test.

20、- L 192-206 You mention using the correlation coefficient (r, not R). What statistical test did you apply to verify the correlation? Spearman, Pearson, Kendall correlation test?

The article uses Pearson's correlation coefficient to verify the correlation and has been added to the original article.

21、 L 192-206 There is no mention of the metrics used to assess the model. Did you compute the R², the RMSE, the %RMSE, the RPD? These metrics are highly relevant and MUST be defined here (principle, formula, interpretation, references).

The metrics for evaluating the model have been added to "method". In this paper, R2 and RMSE are calculated to evaluate the accuracy of the model.

22、-- All the graphical figures (not maps) from this section are unreadable. They MUST appear with the appropriate font size, line color and thickness, figure size and resolution. The manuscript cannot be published with the current figure quality. Important improvements are expected here.

The lines in the image have been bolded and re-exported to replace the previous image.

23、- Table 2 Please add the coefficient of variation for each metal here(CV in %).

The coefficients of variation for each metal have been added to Table 2.

24、- Figure 3 How did you compute the “difference” spectrum. It seems that you subtracted the average contaminated spectrum and the background one. However, there is no mention of that in the Materials and Methods section. Also, I recommend adding the variability (standard-deviation or min-max) for the two average spectra here, in order to better interpret the spectral differences.

Figure 3(b) is to see more intuitively the difference of the original spectral curves under different contamination, we use the spectral curves of the background area minus the spectral curves of the contaminated area, which finally plotted into Figure 3(b). Considering that only simple subtraction is performed here and no complex operations are involved, it is not included in the materials and methods, but we have added the drawing process of Figure 3(b) in the original text. We append the following to the original text: “In order to visualize the difference of the original spectral curves under different contamination, we use the spectral curves of the background area minus the spectral curves of the contaminated area and plot it as Figure 3(b).”

25、- Figure 5 Were the correlations computed using the derivative spectra? If so, please mention it in the figure caption. Also, several heavy metal show the same correlation curve here. How are these metals correlated with each other? For example, if soil Hg and Pb are correlated, there is no need to develop a model for each metal.

Figure 5 uses the original spectral data for the calculation. As seen in Figure 5, the three elements As, Pb, and Cd show the same reflection curve, perhaps due to the existence of certain correlations. However, considering the focus of the article on the correlation between leaf spectra and soil heavy metal content, and the limited space, the relationship between soil heavy metals is relatively unimportant. Then, to ensure the completeness of the article and to continue the previous content, so we modeled all heavy metals.

26、- Figure 6-11 are unreadable and most of these figures are irrelevant. I would recommend showing only the “best” heatmap for each metal, and placing the other figures as appendices.

Figure 6-11 has been corrected to show only the best heat map for each metal, and the complete correlation coefficient distribution has been included as an appendix.

27、- Table 4 Please modify this table so that each metal appear row after row, not as columns (Cr, Hg, and Cu must appear below Cd).

The original table has been revised.

28、- The results presented in section 3.5 must be re-computed using the approach mentioned in a previous comment (train + test and appropriate metrics). These results cannot be published if they are not produced in an appropriate way.

We have added a description of the formulae used in this paragraph to "method", and detailed the method for building the model and testing its accuracy.

29、- Paragraph 3.5.2 This paragraph belongs to the Materials and Methods section.

We have moved the content of the original text that deals with formula methods here to "method".

30、- Paragraph 3.6 This component of the research (mapping) is not appropriate for publication. It is of poor scientific quality and I cannot recommend the article for publication if this “mapping” of heavy metals remains. You apply a regression model developed at leaf scale to 30-m Landsat images with no intermediary step (for example, canopy measurements). Satellite reflectance data are way different than leaf reflectance data. Canopy reflectance as viewed by satellite is influenced by canopy cover density, LAI, soil reflectance, leaf orientation, and many other parameters. Applying a leaf-based model directly to satellite imagery is pure science-fiction. Moreover, you pretend to assess the model’s accuracy on Landsat imagery (L 432-439), but you are linking soil sampled over a few centimeter area to 30-m pixels! I think you should limit you research to leaf-scale measurements here, and remove this non-scientific extension of your research.

We are grateful to the expert teachers for their suggestion. After careful and thoughtful consideration, we found that such an indirect situation is indeed lacking in this paper, so we decided to remove the relevant content of spatial inversion and leave it as a discussion.

31、- This section is poorly written. It is a simple synthesis of the results obtained by the proposed models. There is not comparison with previous work, no discussion about the limitations of the study. This section is too short, too vague, and written in a non-scientific way (the conclusion is longer than the discussion!).

We have integrated the comments of the expert teachers and rewritten the discussion section, and we thank them for their comments and guidance.

Round 2

Reviewer 2 Report

The comments have been addressed and appropriate changes had been made accordingly. I recommend this paper for publication.